# Genus *Ophiorrhiza*: A Review of Its Distribution, Traditional Uses, Phytochemistry, Biological Activities and Propagation

**DOI:** 10.3390/molecules25112611

**Published:** 2020-06-04

**Authors:** Muhammad Taher, Siti Syazwani Shaari, Deny Susanti, Dayar Arbain, Zainul Amiruddin Zakaria

**Affiliations:** 1Department of Pharmaceutical Technology, Kulliyyah of Pharmacy, International Islamic University Malaysia, Kuantan 25200, Malaysia; syazwani1897@gmail.com; 2Department of Chemistry, Kulliyyah of Science, International Islamic University Malaysia, Kuantan 25200, Malaysia; 3Faculty of Pharmacy, Universitas 17 Agustus 1945, Jakarta 14350, Indonesia; 4Department of Biomedical Science, Faculty of Medicine and Health Sciences, Universiti Putra Malaysia, Serdang 43400, Selangor, Malaysia

**Keywords:** genus *Ophiorrhiza*, distribution, traditional uses, bioactive molecules, biological activity, propagation

## Abstract

Almost 50 species of *Ophiorrhiza* plants were reviewed in this work and the main objective is to critically analyse their distribution, phytochemical content, biological activity, and propagation. Moreover, the information would be useful in promoting the relevant uses of the plant, especially in the medicinal fields based on in vitro and in vivo studies. To this end, scientific sources, including theses, PubMed, Google Scholar, International Islamic University Malaysia IIUM EBSCO, PubChem, and Elsevier, were accessed for publications regarding the *Ophiorrhiza* genus in this review. Scientific literature regarding the *Ophiorrhiza* plants revealed their wide distribution across Asia and the neighbouring countries, whereby they were utilised as traditional medicine to treat various diseases. In particular, various active compounds, such as alkaloids, flavonoids, and terpenoids, were reported in the plant. Furthermore, the *Ophiorrhiza* species showed highly diverse biological activities, such as anti-cancer, antiviral, antimicrobial, and more. The genus propagation reported could produce a high quality and quantity of potent anticancer compound, namely camptothecin (CPT). Hence, it is believed that the relevant uses of natural compounds present in the plants can replace the existing crop of synthetic anticancer drugs associated with a multitude of unbearable side effects. Additionally, more future studies on the *Ophiorrhiza* species should be undertaken to establish the links between its traditional uses, active compounds, and pharmacological activities reported.

## 1. Introduction

Genus *Ophiorrhiza* belongs to the Rubiaceae family, which is one of the Indo-Malaysian genera widely spread in the wet forests across tropical and subtropical Asia, Australia, New Guinea, and the Pacific Islands [1,2]. Most of the genera members are perennial herbs capable of growing from approximately 10 cm to 1 metre of height [2]. Normally, the *Ophiorrhiza* genus can be characterised by its succulent stems, five-petal flowers with slightly unequal opposite leaves, laterally compressed fruits, and numerous capsular seeds of small rhomboid shapes [3]. Currently, the *Ophiorrhiza* L. genus consists of 321 species, five varieties, and one subspecies. In particular, 46 species and five varieties are mainly distributed in the north-eastern states and Western Ghats of India [2], whereas 16 species and three varieties can be found in the state of Kerala, India [4].

## 2. Botany

For the past period of years, herbal medicines have been served as therapeutic agents and continue to be important lifesaving drugs for mankind. It is undeniable that plants can treat various forms of diseases including cancer. The demand for herbal drugs increases due to the increased awareness of the side effects and toxicity linked to synthetic drugs [5]. Plants from family Rubiaceae are known to contain several secondary metabolites like alkaloids, flavonoids, steroids, terpenoids, and fatty acids. Some of the compounds from this family, such as caffeine, quinine, emetine, and camptothecin, are of major pharmaceutical importance [6]. These metabolites can be utilized as natural medicines as they can inhibit the activity of DNA topoisomerase, which are the clinical targets for anticancer drugs [7]. Camptothecin (CPT) is a potent anti-cancer compound that has been widely isolated from *Ophiorrhiza* species (*O. mungos, O. mungos var. angustifolia, O. rugosa var. decumbens* and many others), after being first isolated from a Chinese tree, *Camptotheca acuminate,* in 1966 [6,8]. Later, many researchers extensively developed several other CPT analogues, and among them are topotecan and irinotecan, which showed better DNA-topoisomerase-1 inhibitor activity and are well-tolerated compared to natural CPT [9].

## 3. Methodology

In-depth information on the *Ophiorrhiza* genus was obtained via a literature search conducted for publications using various electronic databases, such as Google Scholar, International Islamic University Malaysia (IIUM), EBSCO, PubMed, PubChem, and Elsevier. Accordingly, ChemDraw software was employed to draw the bioactive molecules found in the *Ophiorrhiza* plants. In terms of the publications selected for review, no limitations for the range of years were subjected in this paper. The keyword ‘genus *Ophiorrhiza*’ was used for the primary searches, while the following terms were employed for secondary searches: ‘Plant name’, ‘Phytochemicals’, ‘Biological activity’, ‘Pharmacological activity’, and ‘Propagation’. To highlight the therapeutic uses of genus throughout the medical field, not all of the identified plants were included in this paper. Only those subjected to in vitro and in vivo studies were presented in this review. Therefore, this review was not exhaustive for all *Ophiorrhiza* species typically utilised in traditional or modern medicines. The layout of the searching methodology is presented in Figure 1.

## 4. Distribution

The Western Ghats are known as one of the diversity centres of *Ophiorrhiza* species following the Western Himalayas, which comprise of about 22 distributed taxa for the species [10]. Some of them that can be found in Andaman and the Nicobar islands of India include *O. infundibularis* Balakr, *O. mungos* L, *O. nicobarica* Balakr, and *O. trichocarpa* BL [11]. Meanwhile, about 30–35 species are documented in the shady and moist-to-wet areas in Thailand’s lowland and mountain forests both [3], whereas Peninsular Malaysia is home to 21 species [6]. In contrast, 70 species of the *Ophiorrhiza* are distributed in China in which most of them are found in the southern area of Yangtze River, which traverses the provinces of Yunnan and Guangxi [12]. Figure 2 shows some of the *Ophiorrhiza* collections found in West Sumatra, while Table 1 shows the distribution of this species worldwide.

In general, most of the *Ophiorrhiza* species are widely discovered in the Asian regions, particularly India, Indonesia, and Malaysia. This suggests the highly suitable nature of the regional tropical and forest environments for the growth and distribution of these plants. Therefore, it is believed that newer *Ophiorrhiza* species can be discovered via the undertaking of extensive research in Asian countries.

## 5. Traditional Uses

The plants species of *Ophiorrhiza* (Rubiaceae) genus have been associated to claims of various medicinal properties and wide-ranging applications in traditional and modern medicine alike [4,10,33]. Traditionally, the plants are used to treat inflammation, pain, cancer, and bacterial and viral-based infections. Furthermore, the *Ophiorrhiza* species are capable of healing snakebite, stomatitis, ulcers, and wounds [5,10,33], while also acting as an antioxidant [46], antitussive, and analgesic alternative [4]. They are also applied to tend to cases of gastropathy, leprosy, and amenorrhea, besides possessing sedative and laxative properties obtained from the extract of their root barks [33]. In fact, it interesting to note that *O. mungos* is specifically known as ‘snakeroot’ due to its known use as a treatment for snakebite [6].

In modern medicine, *Ophiorrhiza* plants are popular as a result of their constituent camptothecin’s anti-cancer properties, which are attributable to its ability in inhibiting deoxyribonucleic acid (DNA) topoisomerase-1 [10]. However, their usage in treating different diseases may be dissimilar between varying tribes. For example, the Tanchangya people in Bangladesh uses the paste of *O. rugosa var. prostrata* (D.Don) Deb & Mondal in order to treat boils, those of the Mama tribe make tea from its leaves in tending to body aches or juices them for diarrhoea, whereas the Chakma tribe treats earache by applying sun-dried crushed leaves on the site of pain [36]. Regardless, various alternative traditional uses of this particular species have been detailed, which are shown in Table 2. The *Ophiorrhiza* species are evidently rich with bioactive molecules, offering superior pharmacological effects as they can be used to treat a multitude of diseases throughout mild to chronic stages.

In this review, all available information about *Ophiorrhiza* genus and its bioactive molecules linked with significant pharmacological properties are collected. This act is particularly crucial to promote the plants and their relevant applications in the medicinal fields according to pharmacological evidence obtained via in vitro and in vivo studies.

## 6. Phytochemistry

The Rubiaceae family is known with its capability of producing bioactive metabolites such as iridoids, indole alkaloids, anthraquinones, terpenoids (i.e., diterpenes and triterpenes), flavonoids, and many other derivatives of phenolic compounds, which result in their respective significant pharmacological activities [46]. The phytochemical analyses done for several species of *Ophiorrhiza* (Rubiaceae) have revealed the positive presence of alkaloids, flavonoids, and triterpenes, specifically in *O. radicans* [10], *O. mungos* [5], *O. liukiuensis* [37], and *O. nicobarica* [11]. Furthermore, some of the important phytochemicals found in most *Ophiorrhiza* species include camptothecin and its derivatives, namely pumiloside, luteolin, harman, tetrahydroalastonine, bracteatine, blumeanine, strictosidinic acid, and lyalosidic acid [49]. According to many studies, the therapeutic uses of these compounds can described by their anti-inflammatory, anticancer, antiviral, and antibacterial activities exhibited [46]. Therefore, the compounds derived from *Ophiorrhiza* species are perceived as useful as one of the natural alternative sources in drug development in order to replace chemically-synthesised medications associated with undesirable side effects such as dizziness, nausea, and vomiting.

### 6.1. Alkaloids

The *Ophiorrhiza* (Rubiaceae) species are highly prevalent among plant-rich indole alkaloids [24], which are characterised by a low molecular weight and contain at least one nitrogen atom in an amine-type structure. In their pure form, most of the alkaloids are colourless, non-volatile, and exist as crystalline solids [47]. They are typically useful due to their biological activities and defensive properties observed in plants, as well as for medicinal treatment purposes as they have excellent pharmacological properties [50].

One of the common alkaloids found in *Ophiorrhiza* species is harman (**1**). In particular, harman-2-oxide (**2**) generally found in *O. rosacea* is a crystalline alkaloid that can be reduced to harman (**1**) by reduction with zinc in acetic acid. Furthermore, strictosidinic acid (**3**) is another major bioactive molecule in *Ophiorrhiza* plants besides harman (**1**) [28]. Through Carbon-13 nuclear magnetic resonance (C-NMR), harman (**1**) can be characterised as a compound containing six non-substituted aromatics and one primary and three substituted aromatic carbons, whereas strictosidinic acid (**3**) consists of six quaternary, five methane, and 16 methylene carbons [6].

Next, palicoside (**4**) or *N*-methylstrictosidinic acid depicts a quite similar structure compared to strictosidine (**5**). However, the positioning of additional methyl groups at R^1^ and R^2^ differentiates them into two different structures. In particular, the formation of normalindine (**6**) and isonormalindine (**7**) can be derived from strictosidine (**5**) by introducing the β and α-configurations of a methyl group between N4 and C19, respectively [28]. Similarly, isomalindine-16-carboxylate (**8**) and malindine (**10**) display close spectroscopic data with isomalindine (**9**) at the C3,18 carbons and the N-Me group. However the difference between them is that there is a carboxylate group placed at C6 carbon of isomalindine-16-carboxylate (**8**), while in the case of malindine (**10**), there is a different orientation of the methyl group at C19 as compared to the other two compounds, namely isomalindine (**9**) and isomalindine-16-carboxylate (**8**) [13].

On top of this, camptothecin (**11**) is a modified monoterpene indole alkaloid having potent anti-cancer property has been reported to be found in abundance in *O. mungos* and *O. mungos* var. *angustifolia*, rendering them the main sources for its biomass production [8]. It consists of a pentacyclic ring structure, which includes a pyrrole (3,4β) quinoline moiety and an asymmetric centre within the α-hydroxyl lactone ring with 20S configuration [7]. Additionally, the pumiloside (**12**) and deoxypumiloside (**13**) found in *O. pumila* are further considered as the precursors for the biosynthesis of camptothecin (**11**). This is attributable towards their formation from the half structure of camptothecin (**11**) and half structure of strictosamide (**14**) accordingly [32].

Meanwhile, the two glucoalkaloids of lyalosidic acid (**15**) and 10-hydroxylyalosidic acid (**16**) have been found to coexist with their respective β-carbolines alkaloid counterparts, namely 6-hydroxyharman (**17**) and harman (**1**) in *O. japonica*. This suggests that these two constituents are closely correlated, whereby the hydrolysis of harman (**1**) at C6 will produce 6-hydroxyharman (**17**). Similarly, the diazomethane methylation of lyalosidic acid (**15**) and 10-hydroxylyalosidic (16) acid accordingly yields the two products of lyaloside (**18**) and 10-methoxylyaloside (**19**), respectively [51].

Moreover, the chemical structures of *ophiorrhizine* (**20**) isolated from *O. major* Ridl [25], *ophiorrine* A (**21**), and *ophiorrine* B (**22**) extracted from *O. japonica* [38], and six *ophiorrhisides*, namely *ophiorrhisides* A (**23**), B (**24**), C (**25**), D (**26**), E (**27**), and F (**28**) isolated from *O. trichocarpan* Blume [52] are quite similar. However, they are dissimilar regardless due to the differences in the additional groups and glucose orientation at a certain position of carbon atom. For example, the difference in stereochemistry configuration at C16 will thus differentiate *ophiorrine* A (**21**) and *ophiorrine* B (**22**) into two different betaine-type of indole alkaloids [38]. Accordingly, the chemical structures of the alkaloids under the umbrella of the *Ophiorrhiza* species are displayed in Figure 3.

Furthermore, prior reports have detailed the pharmacological properties of some alkaloids, one of which is harmaline (**29**), also known as harmidine, which is found in *O. nicobarica*. When subjected to oxidation, it will be converted to a psychoactive substance known as harmine (**30**), whereas a reduction process will give rise to d-1,2,3,4-tetrahydroharmine (**31**) [11]. Similarly, compounds such as vallesiaschotamine (**32**), strictosidinic acid (**3**), and tetrahydroalstonine (**33**) have been proven to be capable of depressing the central nervous system, relaxing the muscles, and offering relief pain (i.e., tetrahydroalstonine only) when administered to mice, whereas lyalosidic acid (**15**) yields a vasodilator effect [24].

Various different alkaloids that can be found in *Ophiorrhiza* plants are summarised accordingly in Table 3. The most common types for this particular species consist of harman, ophiorrhine, and camptothecin as well as their respective derivatives. Most of these compounds display a pentacyclic amine-ring structure, which is an indication that they are of the alkaloid group. Therefore, it is undeniable that the *Ophiorrhiza* species is rife and abundant with alkaloid compounds, rendering them plants with a high potential of excellent therapeutic effects.

### 6.2. Flavonoids

Flavonoids are the subgroups of phenolic compounds, which are characterised with at least one aromatic ring and one or more hydroxyl substituents [50]. Their basic structures consist of two benzene rings linked through a heterocyclic pyrane ring [58], whereupon the compounds can be classified into several subclasses, such as isoflavonoids, chalcones, flavanones, flavones, dihydroflavonols, flavonols, anthocyanidins, and catechins [7]. The presence of flavonoids in plants is responsible for providing an attractive colour in order to attract plant pollinators, protecting the leaves from fungal pathogens and UV-B radiation, and controlling normal physiological activities such as respiration and photosynthesis [58]. Furthermore, the compounds are linked to several pharmacological properties, such as anti-inflammatory, anti-platelet, anti-proliferative, anti-carcinogenic, and many others, as well as a minimal toxicity effect [59]. In fact, some of the flavonoids also possess anti-snake venom activity: they occur by inhibiting phospholipase A2, which is an essential constituent in snake venoms [5].

Various plants of the *Ophiorrhiza* species contain flavonoids, whereby *O. mungos* Linn. and *O. liukiuensis* have been specifically reported to have luteolin-7-*O*-glucosides (**48**) and hyperin (also known as quercetin-3-*O*-galactoside) (**49**) respectively (Figure 4). In particular, luteolin-7-*O*-glucosides (**48**) possess three aromatic ring structures with the characteristics of 5,7,3′,4 oxygenated flavones. Hydrolysis of luteolin-7-glycosides with acid will generate luteolin-7-*O*-β-glucosides [60]. Luteolin-7-*O*-glucosides (**48**) typically yield potent antioxidant, anti-inflammatory, and anti-cancer properties [60]. Additionally, it is newly reported that it can protect the gastrointestinal tract from ethanol and indomethacin-induced gastric ulcer upon its modelling in rats [59].

Meanwhile, hyperin (**49**) is a tetrahydroxyflavone compound composed of quercetin and a beta-D-galactosyl residue attached at position 3 [61]. The butanol extract of *O. liukiuensis* whole plant has revealed the presence of hyperin (**49**) and other alkaloid compounds concomitantly [46]. Previously discovered in *Ericaceae, Guttifera,* and *Celastraceae*, the compound has been reported to produce anti-inflammatory effects. For example, it can the inhibit lipopolysaccharide (LPS)-induced inflammatory response by terminating the TLR4 and NLRP3 signalling pathways, which are generally associated with acute kidney injury [61]. This review thus reveals that flavonoid compounds are less abundant and present as much as alkaloids are in the *Ophiorrhiza* species. Regardless, several previous studies have shown that they are capable of providing a significant therapeutic effect to certain diseases in comparison with their alkaloid counterparts.

### 6.3. Terpenoids

Terpenoids, which are also known as isoprenoids, are derived from two precursors, namely isopentenyl diphosphate (IPP) and its isomer dimethyl allydiphosphate (DMAPP). Accordingly, they can be classified into monoterpenes (C_10_), sesquiterpenes (C_15_), diterpenes (C_20_), sesterterpenes (C_25_), triterpenes (C_30_), tetraterpenes (C_40_, carotenoids), and steroids (C_18–30_) [50]. All terpenoids are derived from a repetitive fusion of branched five-carbon units based on the isopentane skeleton [62] and offer varying beneficial uses as flavourings and medicines alike [50].

In particular, ursolic acid (**50**) has been found together with camptothecin (**11**), 10-methoxy-camptothecin, and harman (**1**) from the chloroform extract of *O. liukuensis* whole plant [46]. It is a pentacyclic amphiphilic triterpene compound with a hydroxylated polycyclic structure, which is capable of exhibiting many functions. For example, it has selective induction of cell death properties with the help of caspase-3, may prevent the stimulation of lipoxygenase and cyclooxygenase, and selectively inhibits the cyclic AMP phosphotransferase and AMP-dependent protein kinase. Consequently, the metabolic activity of a cell, as well as its division, gene expression, and development, can be maintained accordingly [11], thereby collectively preventing the cell overgrowth that potentially leads to cancer.

In recent years, new triterpenoid fatty acid esters named lupan-20-ol-3(β)-yl hexadecanoate (**51**), lupan-20-ol-3(β)-yl acetate (**52**), and olean-18-en-3(β)-yl hexadecanoate (**53**) have been isolated from the hexane extract of *O. shendurunii*, which present as a white and waxy solid. These compounds manifest certain antimicrobial, antifungal, and anti-yeast activities through the diffusion disc method. In contrast, other compounds isolated from the hexane extract include doctriacontanoic acid (**54**) and stigmasterol (**55**), whereas rubiadin (**56**), nonadecanoic acid (**57**), hexadecanoic acid (**58**) (i.e., palmitic acid), and camptothecin (**11**) are obtained from the chloroform extract of *O. shendurunii* whole plant [35]. Figure 5 and Table 4 below show the terpenoid content in some of the *Ophiorrhiza* species accordingly.

## 7. Biological Activity

### 7.1. Anticancer Activity

In general, normal cells will grow and divide themselves for some time before their growth stops, whereas reproduction will only occur in order to replace any damaged or dead cells. Therefore, cancer occurs in the presence of abnormal cell growth due to the altered gene expression of DNA in the affected cells, consequently resulting in their resistance to apoptosis and continuous division [63]. Moreover, the existence of reactive oxygen species (ROS) may further promote the development of these tumour cells as they directly alternate the cellular metabolic process. If the natural antioxidants found in the body cannot control the production of ROS or destroy them, severe damage to body tissues may be observed while the cancer cells continue to actively proliferate [60].

The conventional synthetic anticancer drugs have unbearable side effects such as fatigue, sleep disturbances, and hair loss, thereby rendering their substitution slowly with nature-derived alternatives [63]. In particular, the therapeutic effects of secondary metabolites possessed by alkaloids, flavonoids, terpenoids, polyphenols, and quinones may potentially inhibit the activity of DNA topoisomerase enzyme [62] and suppress the ROS activity, thus leading to fewer side effects [60].

The actions of the DNA topoisomerase enzyme causes DNA strands to be separated and re-joined, resulting in changes of the structure [62]. Accordingly, camptothecin (**11**) is one of the powerful alkaloids found in *Ophiorrhiza* plants in possession of excellent anticancer activity and acts as a potent inhibitor of DNA topoisomerase I [10,64]. The component is capable of preventing the enzymatic activity of DNA topoisomerase I specifically by trapping the reaction intermediate (i.e., the cleavable complex) during the breaking and re-joining process, causing disrupted cancer cell division [62]. Furthermore, it can bind to tubulin and inhibit microtubule formation in the dividing cells, subsequently enhancing the inhibition of the DNA topoisomerase [64]. Therefore, this review offers the notion that the development of novel anticancer drugs extracted from natural sources is possible due to the pharmacological effects of unique compounds present in the *Ophiorrhiza* plants.

Besides camptothecin (**11**), a multitude of phytochemicals show great anticancer properties as well, such as harmane (**1**) and harmaline (**29**)’s reported roles in scavenging the ROS and inhibiting thiol oxidation [65]. Accordingly, a comparative study has noted luteolin and quercetin’s ability to inhibit topoisomerase II catalytic activity in Chinese hamster ovary AA8 cells, underlining them as a pioneering lead for cancer treatment [7]. Meanwhile, the UV filtering property of 11 natural plants including *O. mungos* has also been assessed in a preliminary study. Following this, the results suggest that secondary metabolites such as polyphenols and flavonoids can absorb UV radiation and scavenge the ROS produced from body cells [66]. This is an essential outcome as the free reactive atoms or molecules are harmful due to their ability to damage normal and key biological macromolecules in the body, such as lipid, protein, and DNA. In contrast, the anthraquinone fraction of *O. rugosa van decumbens* functions by causing cytotoxicity in cancer cells and increasing the intracellular ROS, thus leading to cell apoptosis [67].

Several works assessing the anti-inflammatory activities of the extracts obtained from *Ophiorrhiza* species have been conducted as they are highly beneficial in determining their respective anticancer properties. Inflammation is typically complicated, beyond one’s control, and a continuous process that results in the progression of diseases such as atherosclerosis and cancer. Therefore, carrageen-induced inflammation is a commonly employed method in order to investigate the anti-inflammatory activity of certain compounds. To date, no exact mechanism of action for the anti-inflammatory activity of anthraquinone fraction from *O. rugosa* has been underlined. However, it is believed that inflammation reduction occurs via the inhibition of the lipid peroxidation and scavenging of hydroxyl radicals [67]. Table 5 shows the anticancer activities of certain *Ophiorrhiza* species based on previous studies.

### 7.2. Camptothecin Interest

Camptothecin (**11**) was first isolated via the extraction of the bark and stem of the Chinese tree *Camptotheca acuminata* in 1958 [8,69]. It was later found in *C. lowreyana, C. yunnanensis, Nothapodytes nimmoniana, Pyrenacantha klaineana, Merrilliodendron megacarpum, Ervatamia heyneana, Mostuea brunonis, O. mungos, O. pumila,* and *O. filistipula* as well [4,10,34,70]. Even though camptothecin is one of the potent topoisomerase-1 inhibitors, its usage and application in clinical trials and global market has been withdrawn as a result of its low aqueous solubility and severe toxicity during the *S*-phase of the cell cycle, which can lead to normal cell death [9]. In fact, the early 1970s yielded various camptothecin clinical trials in which the respective researchers had failed across several attempts in developing its sodium salt form to overcome its poor solubility characteristics. Following this, they had found that for camptothecin (**11**) to show its activity, the lactone ring moiety must remain intact. However, the ring was opened during the sodium salt preparation, thus resulting in its failure to show the anticipated activity [71].

Another earlier study has also suggested that a complete pentacyclic ring structure with the D ring pyrridone and E ring lactone formed via a 20S configuration is crucial towards the anticancer property of camptothecin (**11**) [7]. Scholarly interest in camptothecin molecules steadily increased in the mid-1980s in which two first-generation water-soluble camptothecin analogues with an intact lactone ring were developed, namely camptosar (irinotecan or CPT-11) (**62**) and hycamtin (topotecan) (**63**) (Figure 6). These two structures were obtained by modifying the A and B rings of camptothecin accordingly [7]. In 1996, the United State Food and Drug Administration (FDA) approved the use of these two analogues in the pharmaceutical industry, whereby they were marketed by Pharmacia (Pfizer) and GlaxoSmithKline for the treatment of metastatic colorectal, primary colon, and metastatic ovarian cancers [71].

Regardless, recent studies undertaken have shown that the isolated natural camptothecin derivatives such as 9-methoxycamptothecin and 10-hydroxy-camptothecin are water-soluble and possess antitumor activity [70]. Following this, more advanced therapeutic strategies by using target-oriented drug delivery systems have crowned camptothecin (**11**) as a major candidate for the treatment of multi-drug resistant cancers [8]. In fact, newer drug delivery via liposomal and copolymer vehicle-mediated systems are capable of significantly improving the safety and efficacy of CPT [9]. For example, its release mediated by pH-sensitive block copolymer is believed to enhance its subsequent delivery in the body, alongside better safety considerations [72]. Additionally, another new alternative strategy of stem cell-based research in the apoptogenic signalling of CPT is still in progress [73].

From these findings, clearly several approaches have been discovered by researchers in order to stimulate the application of CPT in regenerative medicine by using its chemical analogues. Among them, methods such as manipulating its ring structure, conjugating it with other molecules, developing its prodrug, and using specific protein-based drug delivery systems have been suggested to establish a better efficacy and safety of anticancer drugs. However, most of the analogues are still in the phase of clinical trials and their safety issues are attributed as the major reason why these compounds have yet to enter the market. Table 6 shows the camptothecin analogues that are still in the clinical trial phase in detail.

### 7.3. Antiviral Activity

Generally, flavonoids are one of the phytochemicals responsible for the activities against multiple viruses. For example, catechin can inhibit the property of viral infectivity of the respiratory syncytial virus (RSV) and herpes simplex virus-1 (HSV-1) but not their intracellular replication, whereas quercetin is capable of effectively reducing the infectivity of majority of viruses [74] and partially inhibiting DNA gyrase [58]. Meanwhile, coumarins may possess antiviral effects by stimulating the macrophages towards preventing infections and recurrence of cold sores caused by HSV-1 in humans, while terpenoids are active against yeasts such as *Candida albicans.* Similarly, harmane offers intercalating properties with the DNA of viruses [74] and has a strong activity against the intracellular amastigote of *Leishmania infantum,* which is partly due to its ability to inhibit the leishmanial protein kinase C (PKC) activity [75].

Furthermore, harmaline (**29**) or 7-methoxy-1-methyl-4,9-dihydro-3*H*-pyrido[3,4-β] indole alkaloid isolated from *O. nicobarica* is known for its potent antiviral activity against herpes simplex virus 1 (HSV-1. Rather than interfering with viral entry, it affects the recruitment of lysine-specific demethylase-1 (LSD1) and immediate-early (IE) complex binding on the promoter of ICP0. This will, in turn, contribute to the suppression of the viral IE gene synthesis, thereby causing a reduction of ICP4 and ICP27 expression [76]. Similarly, camptothecin (**11**) and 10-methoxycamptothecin previously isolated from the leaves of *O. mungos* have been noted to yield an active activity against the herpes virus [77]. Table 7 shows the antifungal, anti-yeast, and antiviral activities of five *Ophiorrhiza* species according to prior studies conducted on these plants.

### 7.4. Antimicrobial Activity

The components of alkaloids, carbohydrates, tannins, anthraquinones, hormones, terpenoids, saponins, flavonoids, essential oils, glycosides, and potassium are undeniably considered to have a curative effect against several pathogens [78]. Flavonoids, for example, can form a complex with extracellular bacterial soluble proteins and cell walls; those that are lipid-soluble may also disrupt the microbial membranes as they are lipophilic in nature. Meanwhile, coumarins and terpenoids are active against positive bacteria such as *Bacillus subtilis* and *Staphylococcus aureus,* despite the latter’s lesser effect against gram-negative bacteria [74].

A majority of works assessing the antimicrobial activities of extracts from *Ophiorrhiza* species have been conducted using the diffusion assay method, namely by observing the inhibition zone of these extracts on certain microorganism growth. Different plant extracts will typically display distinct antibacterial activities, which are attributable to their varying phytochemical contents and properties. For instance, the ethyl acetate and methanol extracts have been underlined as the best options amongst other solvent extracts tested. Here, the lesser activity of other extracts is due to their inability to either extract the bioactive molecules with antimicrobial constituents or produce an effective concentration of antibacterial constituents [79]. Table 8 shows the antimicrobial activities of five *Ophiorrhiza* species, whereas their biological activities are summarised in Figure 7.

## 8. Propagation

Over time, several researchers have conducted biotechnology research on certain species of *Ophiorrhiza* in order to develop an efficient method for the development of camptothecin (**11**) and recognition of novel secondary metabolites from cultivated plants [32]. The high demand for camptothecin and other active pharmaceutical ingredients, as well as the overexploitation of *C. accumata* and *Nothapodytes foetida* as the main sources of this component have resulted in the development of various protocols for its in vitro cultivation and production by using *Ophiorrhiza* plants [1]. In fact, camptothecin is a highly valuable drug with market prices ranging from US$3500 to US$350,000 per kilogram. Therefore, it is considerably interesting for scholars to identify alternative ways of producing the component, such as by using plant cells and tissue cultures, as a result of its commercial value [64].

The successful regeneration of *O. pumila* from the callus tissue of its leaves and shoot has been reported, whereby a new glucosyloxy camptothecin (i.e., 9-β-glucosyloxycamptothecin) is isolated along with 15 metabolites, including six alkaloids linked to camptothecin (**11**). However, (3*S*)-deoxyppumiloside as one of camptothecin’s possible biogenetic precursors has not been detected, whereas the (3*R*) epimer is isolated from regenerated plants. Accordingly, these well-developed callus cultures can also generate the anthraquinones which are not producible by non- culture *O. pumila* plants [54]. Therefore, a method for the micropropagation and generation of camptothecin from *O. mungos* via in vitro plants has been documented, whereas a protocol is also developed for rapid root and organogenesis proliferation. As a result, the high performance liquid chromatography (HPLC) analysis of the method has proven that its in vitro production generates higher yields than natural-grown plants [80].

Moreover, a study on the effect of jasmonic acid in the cell suspension culture of *O. mungos* species has revealed the suspension’s ability to increase the production of camptothecin significantly [81]. Similarly, another study has substantiated the effect of silver nitrate and yeast extract on the cell growth, camptothecin (**11**) accumulation, and cell viability, thereby leading to a significant increment in the production of biomass and camptothecin (**11**) alike [82]. In fact, a prolonged subculture of *O. trichocarpose* Blume by alternating the medium intensity in each subculture has resulted in a notable increment in the development of the plant’s shoots and biomass [83].

In line with this, different concentrations of various auxins in *O. mungos var. angustifolia* culture have also been reported to influence the biomass camptothecin (**11**) production, whereby the result shows that both explants will not induce callus growth in the absence of exogenous hormones. In contrast, they will induce the growth of calli following their culture in the Murashige and Skoog (MS) medium at different strengths together with butyric acid (BA) and naphthalene acetic acid (NAA) solutions. The study underlined the ideal combination of solutions as half-strength MS solid medium with 10.74 µM NAA + 4.44 µM BA. Further, the scholars have discovered that the in vitro leaves are the best explants that can help to produce a higher content of camptothecin compared to their in vivo counterparts [69].

In contrast, an efficient plant regeneration system has been introduced in a study through somatic embryogenesis. Here, the embryogenic callus tissues extracted from aseptic leaf explants were used to induce the somatic embryos under callus tissues incubated across the different periods of 10 to 60 days. The outcomes showed that 40-day-long incubation in 0.5 mg/L 2,4-dichloro phenoxy acetic acid media was the optimum environment for the somatic embryogenesis in *O. pectinata.* Therefore, the study successfully substantiated the influence of callus tissue incubation period towards the differentiation of embryos in suspension cultures [33].

Moreover, a high production of camptothecin (**11**) has been obtained using the root cultures of both transformed and untransformed *O. mungos* [64] and *O. alata Craib* [39], respectively, thus promoting an alternative novel system in sustaining its bio-production as raw material for the pharmaceutical industry. The transformed hairy roots were obtained by infecting the node explants with *Agrobacterium rhizogenes* TISTR 1450 [39,64]: it was found that the CPT (**11**) concentration was consequently double the amount in the soil-growing plant [39].

In general, culture media containing ammonium as the nitrogen source is preferable, whereby nitrate is essential in regulating the pH value and promoting the growth of root and synthesis of camptothecin (**11**) [64]. The addition of polystyrene resin or Diaion HP-20 capable of absorbing camptothecin (**11**) has further led to its increased concentration in culture media, namely seven-fold higher compared to the control media [39]. Moreover, camptothecin (**11**) can be extracted more in root and hairy root cultures as they consist of differentiated tissues. Here, the secondary plant metabolite biosynthesis is still active and has not decreased in contrast to the cultures of undifferentiated callus or cell suspension [39,64].

Nevertheless, the accumulation of camptothecin (**11**) in the different parts of plants is identical [84], rendering it crucial for one to consider the specific selection and isolation of cells or organ lines in order to induce in vitro calli in biomass metabolite production. For example, undifferentiated callus and suspension cultures do not always produce the desired amount of compound of interest. In contrast, the shoot, root, and hairy root cultures will often generate the same compounds as in the appropriate organs [67]. Hence, the type of medium culture, suitable salt strength, nitrate level, phosphate level, and growth regulator level should be considered beforehand to obtain the desired metabolites at a high quantity and quality [84].

## 9. Conclusions

To conclude, compounds derived from *Ophiorrhiza* plants play an important role in treating diseases although they may not directly consider drugs. The compounds can be served as lead compounds that are beneficial for the development of potential anticancer drugs, particularly. This may help new researchers in understanding the diseases better, providing more efficient therapies with a new mechanism of action, increasing the patient compliance, reducing the synthetic anticancer drugs-related adverse effects as well as encouraging the development of future and novel anticancer drugs. The *Ophiorrhiza* species represent an enormous diversity throughout the world, yet not many have been explored. With the development of technology of apparatus and equipment nowadays, newly discovered bioactive compounds showing inhibition effects on cancer cells may rapidly be recognized. Thus, further extensive research and studies should be continued to discover more bioactive molecules in this species and their therapeutic effects to highlight the use of natural source-derived drugs in the medicinal field which are believed to be more well-tolerated and cost-effective.

## Figures and Tables

**Figure 1 molecules-25-02611-f001:**
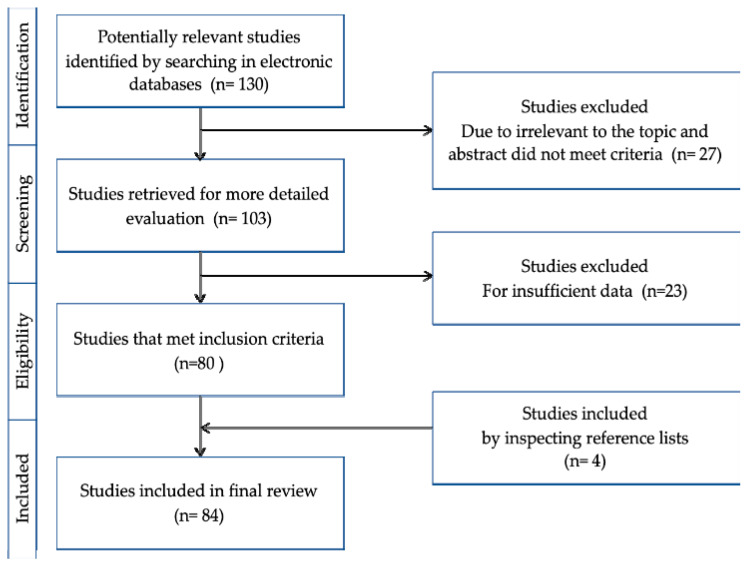
Methodology conducted.

**Figure 2 molecules-25-02611-f002:**
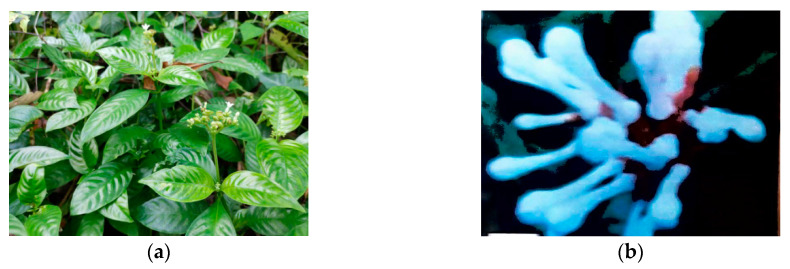
Photo collections of *Ophiorrhiza* species. (**a**) *Ophiorrhiza communis* [13]. (**b**) *Ophiorrhiza* sp. (*ex.* Gunung Singgalang) [14]. (**c**) *Ophiorrhiza* “Siberida DA-RT 6030” [15]. (**d**) *Ophiorrhiza * sp. (ex. Sako, TNKS) [16]. (**e**) *Ophiorrhiza longiflora* BI. [17]. (**f**) *Ophiorrhiza* cf. *kunstlery* King. [18]. (**g**) *Ophiorrhiza* ex. Padang Panjang [19,20]. (**h**) *Ophiorrhiza* “DA-RT 7895” [21]. (**i**) *Ophiorrhiza* “Air Sirah DA-RT 6604” [22]. (**j**) *Ophiorrhiza* “Sako DA-RT 7577” [16].

**Figure 3 molecules-25-02611-f003:**
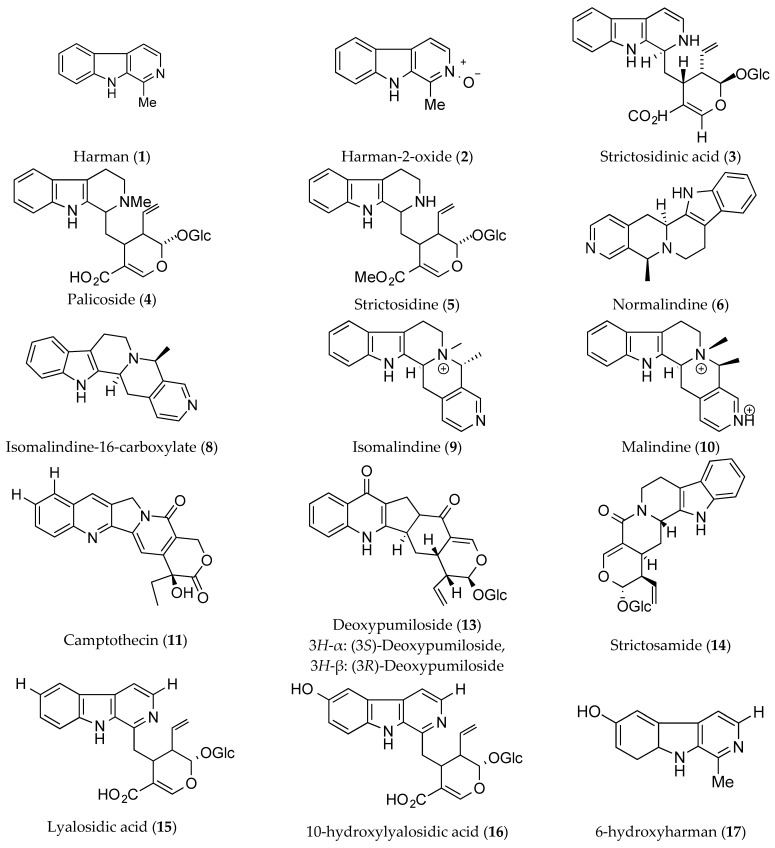
Alkaloids from the genus of *Ophiorrhiza.* *Glc = glucose molecule.

**Figure 4 molecules-25-02611-f004:**
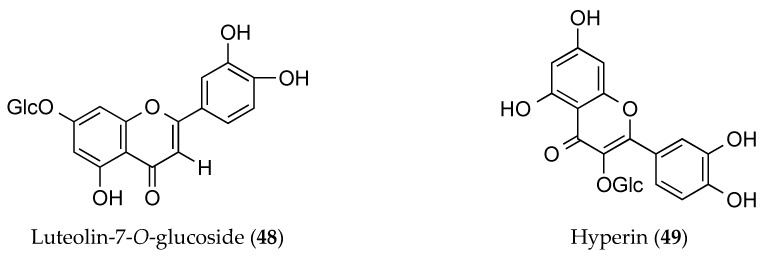
Flavonoids from the genus of *Ophiorrhiza.* *Glc = glucose molecule.

**Figure 5 molecules-25-02611-f005:**
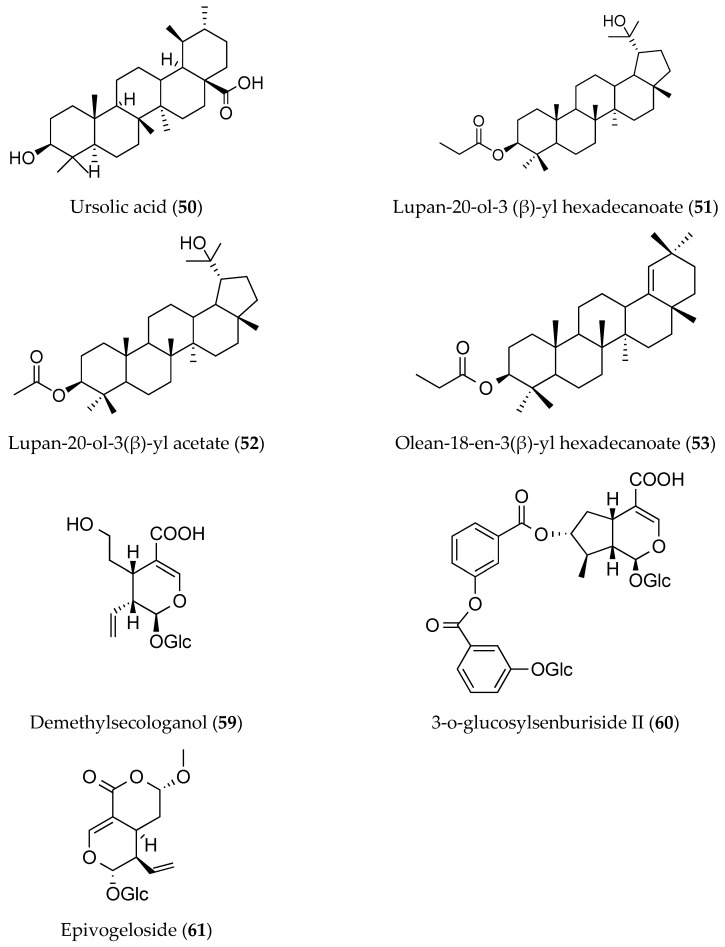
Terpenoids from the genus of *Ophiorrhiza.* *Glc = glucose molecule.

**Figure 6 molecules-25-02611-f006:**
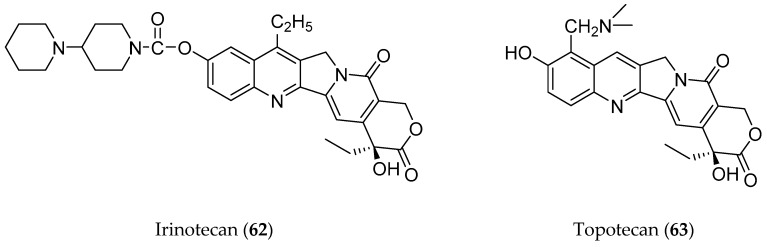
Campthotechin analogues.

**Figure 7 molecules-25-02611-f007:**
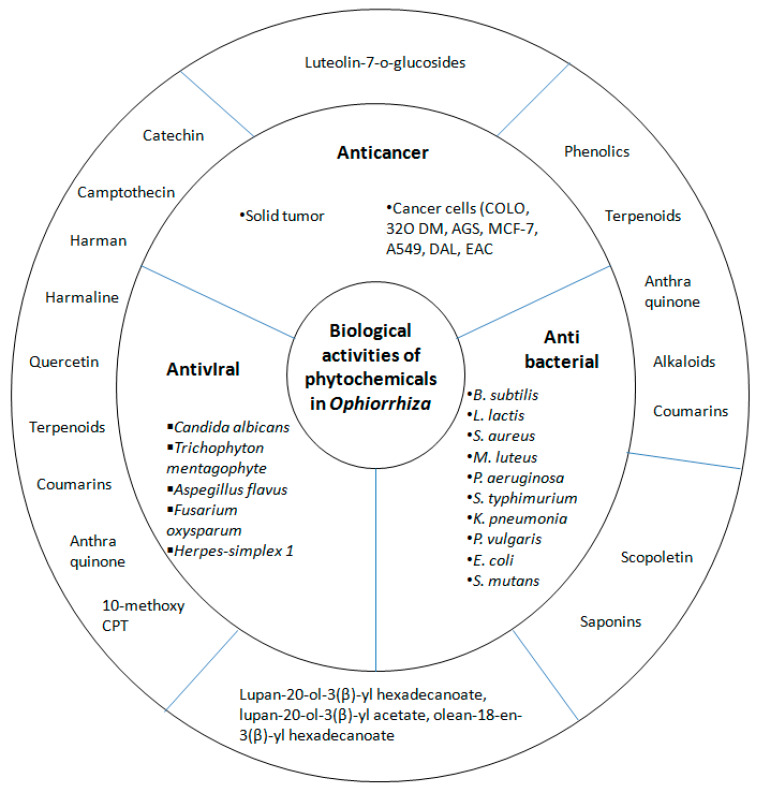
Summary of biological activity of phytochemicals in *Ophiorrhiza* species.

**Table 1 molecules-25-02611-t001:** Distribution of *Ophiorrhiza* species around the world.

Species	Location	References
*O. marginata*	Mt Keronsong, Jambi, Indonesia	[23]
*O.* cf. *rosacea*	Solok, West Sumatra	[24]
*O. communis*	Ulu tembeling Malaysia Bungus, Padang, Indonesia	[6]
*O.* cf. *communis*	Sijunjung, West Sumatra	[13]
*O. tomentosa*	Cameron Highland, Malaysia	[6]
*O. bracteata*	Lubuk Alung West Sumatra	[25]
*O. major*	Anai Reserved Forest, West Sumatra	[26]
*O.* cf *ferruginea*	Anai forest, West Sumatra	[27]
*O. rosacea*	The Seven Mountain, Jambi	[28]
*O. kunstlery*	Pariaman, West Sumatra	[28]
*O.* cf. *kunstlery* King	Mount Letter W Kabupaten Padang Pariaman, Indonesia	[29]
*O. filistipula*	Pangian Lintau, West Sumatra	[30]
*O. blumeana*	Mount Tandikat, West Sumatra	[31]
*O. trichocarpon* Blume	Kerala, South India	[4]
*O. pumila*	Japan	[32]
*O. pectinata*	Tamil Nadu Kerala, India	[33]
*O. radicans*	Western Ghats Kerala, India	[34]
*O. shendurunii*	Kerala Western Ghast, India	[35]
*O. prostrata*	Agastyamala Hills of Western Ghats Thiruvananthapuram Kerala, India	[1]
*O. rugosa*	The northern part of Thailand	[3]
*O. rugosa var. prostrata (D.Don) Deb & Mondal*	Chittagong Hill Tract and Sylhet regions of Bangladesh	[36]
*O. liukiunseis*	Ishigaki Island of Okinawa, Japan, Taiwan, Philipines	[37]
*O. kuroiwai*	Ishigaki and other south-west islands of Okinawa	[38]
*O. japonica*	Ishigaki and other south-west islands of Okinawa	[38]
*O. alata* Craib	Chantaburi, Thailand	[39]
*O. nicobarica* Balakr.	Andaman and Nicobar Islands, India	[11]
*O. infundibularis*	Great Nicobar Island, India	[2]
*O. jojui*	Andaman Islands, India	[2]
*O. shiqianensis*	Guizhou Province, South-western China	[12]
*O. neglecta* BI. ex DC	Kabupaten 50 Kota, Indonesia	[40]
*O. longiflora* BL.	Tarok Lubuk Bonta, Indonesia	[17]
*O. palidulla* Ridl.	Tilatang Kamang, Indonesia	[41]
*O.* sp.	Kotamadya Padang Panjang, Indonesia	[19,20]
*O. klosii* Ridl.	Kabupaten Solok, Indonesia	[42]
*O.* ex. Simanau DA-RT 61, Solok	Kabupaten Solok, Indonesia	[43]
*O. neglecta* BI. Ex DC	Kabupaten 50 Kota, Indonesia	[40]
*O. anonyma* Val.	Silayang Kabupaten Agam, Indonesia	[44]
*O.* ex. Gunung Singgalang	Mount Singgalang Kabupaten Agam, Indonesia	[14]
*O.* “Air Sirah DA-RT 6604”	Air Sirah Kabupaten Solok Sumatera Barat, Indonesia	[22]
*O.* “Sako DA-RT 7577”	Sako Kabupaten Sumatera Barat, Indonesia	[16]
*O.* “Siberida DA-RT 6030”	National Park Siberida Rengat Riau, Indonesia	[15]
*O.* DA-RT 82 AT	Tes Lake Bengkulu, Indonesia	[45]
*O.* DA-RT 7895	Kerinci Seblat National Park (TSNK) Jambi, Indonesia	[21]

**Table 2 molecules-25-02611-t002:** Traditional uses of *Ophiorrhiza* species.

*Ophiorrhiza* Species	Traditional Uses	Hippocratic Screening	References
*O. discolor* Br.	Skin infections		[47]
*O. filistipula* Miq.	Skin infections and inflammation		[47]
*O.* cf. *rosacea* Ridl.		Vasodilator effect	[47]
*Ophiorrhiza* DART 6526		Analgesic and muscle relaxant	[47]
*O. major* Ridl.	Skin disorders like eczema		[25]
*O. nicobarica*	Herpetic lesions, skin infections, and irritation		[11]
*O. mungos* (root)	Cancer and snakebite, Sedative and laxative properties		[48]
*O. communis*	Poultice and treating cough		[6]
*O. tomentosa*	Poultice		[6]
*O. rugosa var. prostrata* (D. Don) & Mondal	Skin infections such as boils, body aches, and chest pain, earache, dysentery, diarrhea		[36]
*O. singaporiensis*	Snakebite		[17]

**Table 3 molecules-25-02611-t003:** Alkaloids contents in *Ophiorrhiza* species.

Species	Alkaloids Contents	Parts of Plant	References
*O. rosacea*	Harman (**1**), Harman-2-oxide (**2**)	Aerial part	[28]
*O. kunstlery* King	Palicoside (**4**), isomalindine (**9**), 3,14-didehydro-19-methylnormalindine (**34**)	Aerial part	[18,28,29]
*O. communis*	Harman (**1**)	Leaves	[53]
*O. tomentosa*	Harman (**1**), strictosidinic acid (**3**)	Leaves	[53]
*O. japonica*	Lyalosidic acid (**15**), 10-hydroxylyalosidic acid (**16**), 6-hydroxyharman (**17**)	Aerial part	[51]
*O. japonica*	Ophiorrine A (**21**), Ophiorrine B (**22**)	Leaves	[38]
*O. kuroiwai* Mak	Ophiorrine A (**21**), Ophiorrine B (**22**), lyasoside (**18**)	Aerial part	[38]
*O. marginata*	Decarbomethoxydihydrogambirtannine (**35**), strictosidinic acid (**3**)	Aerial part	[23]
*O.* cf. *communis*	Isomalindine-16-carboxylate (**8**)	Aerial part	[13]
*O. liukiuensis*	Camptothecin (**11**), 6-hydroxyharman (**17**)	Whole plant	[37]
*O. pumila*	Camptothecin (**11**), chaboside (**36**), pumiloside (**12**), deoxypumilosides (**13**)	Whole plant	[54]
*O. nicobarica*	Harmaline (**29**)	Whole plant	[11]
*O.* DA-RT 6526	Tetrahydroalstonine (**33**), Vallesiaschotamine (**32**), Isovallesiachotamine (**37**)	Aerial part	[47,55]
*Ophiorrhiza* DA-RT 6526b	Strictosidinic acid (**3**)	Aerial part	[24,56]
*Ophiorrhiza* DA-RT 82 A	Tetrahydroalstonine (**33**)	Aerial part	[45]
*O. longiflora* BL.	Tetrahydroalstonine (**33**)	Aerial part	[17]
*O.* cf. *rosacea*	Harman (**1**), lyalosidic acid (**15**)	Aerial part	[24]
*O. mungos*	Camptothecin (**11**)	Leaves	[57]
*O. blumeana* Korth	Ophiorrhizine-12-carboxylate (**38**), bracteatine (**39**), blumeanine (**40**)	Aerial part	[47]
*O. bracteata* Korth	Ophiorrhizine (**20**), bracteatine (**39**)	Aerial part	[47]
*O. discolor* Br.	Tetrahydroalstonine (**33**)	Aerial part	[47]
*O. filistipula* Miq.	7-methoxy-camptothecin (**41**), normalindine (**6**), strictosidinic acid (**3**)	Aerial part	[47]
*O.* cf. *ferruginea* Valeton	Dihydrocycloakagerine (**42**), mostuenein (**43**), tetrahydroakagerine (**44**), isomalindine (**9**)	Aerial part	[47]
*O. major* BI.	Ophiorrhizine (**20**)	Aerial part	[47]
*O. teysmaniana* Miq.	Tetrahydroalstonine (**33**)	Aerial part	[47]
*O. trichocarpon* Blume	Ophiorrhiside A (**23**), Ophiorrhiside B (**24**), Ophiorrhiside C (**25**), Ophiorrhiside D (**26**), Ophiorrhiside E (**27**), Ophiorrhiside F (**28**), dolichantoside (**45**), 5-carboxystrictosidine (**46**), lyaloside (**18**), 3,4,5,6-Tetrahydrodolichantoside (**47**)	Whole part	[52]

**Table 4 molecules-25-02611-t004:** Terpenoids content in *Ophiorrhiza* species.

Species	Triterpenes	Parts of Plants	References
*O. liukiuensis*	Ursolic acid (**50**), demethylsecologanol (**59**), 3′′′-o-glucosylsenburiside II (**60**) (monoterpene), epivogeloside (**61**),	Whole plant	[37,46]
*O. nicobarica*	Ursolic acid (**50**)	Whole plant	[11]
*O. shendurunii*	Lupan-20-ol-3 (β)-yl hexadecanoate (**51**), lupan-20-ol-3(β)-yl acetate (**52**), olean-18-en-3(β)-yl hexadecanoate (**53**)	Whole plant	[8]

**Table 5 molecules-25-02611-t005:** Anticancer activity.

Species	Phytochemicals	Extraction and Isolation	Biological Activity	Method (Dose/ Concentration)	References
*Ophiorrhiza mungos* Linn.	Luteolin-7-*O*-glucosides (**48**)		Gastroprotective effect against ethanol-induced and indomethacin-induced gastric injury in rats	25 mg/kg b.w p.o	[59]
		Alcohol and aqueous extracts of *O. mungos* leaves	Anti-cancer property on Dalton’s Ascites Lymphoma (DAL) in mice	Alcohol extract: 400 mg/kg b.w p.o (14 days) Aqueous extract: 800 mg/kg b.w. p.o (14 days)	[68]
	Luteolin-7-*O*-glucoside (**48**), camptothecin (**11**)	Methanolic extract of leaves and roots of the plant	Anti-cancer activity against COLO 320 DM, AGS, MCF-7, and A549 cancer cell lines in rats Notes: COLO 320 DM (human colon adenocarcinoma), AGS (human gastric cancer cell line), MCF-7 (human breast cancer cell line), A549 (human lung cancer cell lines) Suppressed β-catenin in COLO 320 DM	20 mg/kg b. w subcutaneously for 16 weeks and 120μM (Incubated for 24 h)	[60]
		Methanolic extract of leaves	UV filtering potential	1 mg/mL (The extract was exposed to direct sunlight for 21 days)	[66]
*Ophiorrhiza rugosa van decumbens*	Anthraquinones		Anti-cancer property in mice with: Ehrlich ascites carcinoma (EAC) Dalton’s lymphoma ascites (DAL) Solid tumors	130 µg/mL b.w 160 µg/mL b.w 200 mg/kg b.w (p.o for 10 days)	[67]
*Ophiorrhoza nicobarica*	Major chemicals: ursolic acid (**50**), β-sitosterol, harmaline (**29**)	Alcoholic extract	Analgesic and anti-inflammatory activity in rats and mice	200 and 300 mg/kg b.w p.o Fractions: 50 mg/kg b.w p.o	[11]
***Ophiorrhiza mungos***	Presence of flavonoids, cardiac glycosides and phenolics		Prevention of hemorrhagic lesion induced by snake venom on the yolk sac membrane of the chick embryo.	10 µg/µL (Incubation of the venom with the extract before applying to the embryo for 30 min)	[5]

b.w = body weight; p.o = oral route.

**Table 6 molecules-25-02611-t006:** Clinical trials of camptothecin analogues. Adapted from [9].

Camptothecin Analogues	Year Started	Clinical Trials/Progression	Indication	Company
9-aminocamptothecin	1993	Phase I/II	Ovarian and malignant lymphoma	
Karenitecin (BNP-1350)		Phase II	Malignant melanoma and brain tumor	
Diflomotecan (BN-80915)	2007	Phase II	Solid tumors and small cell lung cancer (SCLC)	Ipsen
Gimatecan (ST-1481)		Phase II	Advanced solid tumors and recurrent epithelial ovarian and fallopian tube cancers	
Elomotecan (BN-80927)		Phase I	Advanced solid tumors	Ipsen and Roche
DRF-1042		Phase I		Dr. Reddys Laboratories
Exatecan mesylate		Phase II	Gastric cancer and relapsed rhabdomyosarcoma 9 in children)	
Rubitecan		Phase II/III but already withdrawn due to unfavorable results	Pancreatic cancer	
CZ-48 *		Phase I	Solid tumors	
TP-300 *		Phase I	Advanced solid tumors	
EZN-2208 *		Phase I	Advanced malignancy	
MAG-CPT **		Discontinue after Phase I		Pharmacia and Upjohn
XMT-1001 **		Phase II	Lung cancer	
CRLX-101 **		Phase I/II	Advanced solid tumor	

* Conjugated form of camptothecin analogue; ** Prodrug form of camptothecin.

**Table 7 molecules-25-02611-t007:** Antiviral activity.

Species	Phytochemicals	Extraction and Isolation	Biological Activity	Method (Dose/ Concentration)	References
*Ophiorrhiza trichocarpon* Bl., *Ophiorrhiza rugosa*, and *Ophiorrhiza aff*. nutans Cl. ex Hk. f.	Alkaloid, coumarin, anthraquinone glycoside, scopoletin, saponins	Methanolic extract of the whole plant	Antifungal activity was determined by the agar-well diffusion method. Inhibition zone: *Candida albicans* (10 mm) *Trichophyton mentagophyte* (12 mm) *Aspergillus flavus* (12 mm)	50 mg/mL 30 mg/mL 75 mg/mL	[3]
*Ophiorrhiza shendurunii*	Lupan-20-ol-3(β)-yl hexadecanoate (**51**), lupan-20-ol-3(β)-yl acetate (**52**), olean-18-en-3(β)-yl hexadecanoate (**53**)	Hexane and chloroform extract of the whole plant	Anti-yeast/antifungal activity of: The three compounds against: *C. albicans* (inhibition zone; 15, 10 and 16 mm respectively) *Fusarium oxysporum* (inhibition zone; 10, 8 and 8 mm respectively). Hexane extract: *Candida albican* (28/28 mm) *Fusarium oxysporum* (18/26 mm) Chloroform extract: *Candida albican* (28/28 mm) *Fusarium oxysporum* (24/26 mm)	100 µL	[35]
*Ophiorrhiza nicobarica*	harmaline (**29**)		Anti-HSV-1 activity in mice	0.25 and 0.5 mg/kg b.w p.o once daily (8 days)	[76]

**Table 8 molecules-25-02611-t008:** Antimicrobial activity.

Species	Phytochemicals	Extraction and Isolation	Biological Activity	Method (Dose/ Concentration)	References
*Ophiorrhiza mungos*	Presence of carbohydrates, proteins, phenolics, terpenoids, glycosides, reducing compounds and saponins	Hexane, chloroform, ethyl acetate, methanol and ethanol extracts of leaves and stem of the *O. mungos.*	Ethyl acetate and methanol extract showed potent antimicrobial activity against nine microorganisms (*B. subtilis, L. lactis, S. aureus, M. luteus, P. aeruginosa, S. typhimurium, K. pneumonia, P. vulgaris, E. coli.).* Inhibition zone: Ethyl acetate extract: 19–28 mm Methanol extract: 11–26 mm	500 µg/mL	[79]
	Carbohydrate, tannin, terpenoids, saponins, flavonoids, alkaloids and glycosides	Ethanolic extract of fresh flower	Antimicrobial activity against six pathogens with inhibition zone: *B. subtilis* (20 mm), *K. pneumonia* (19 mm)*, P. aeruginosa* (14 mm)*, S. aureus* (14 mm)*, S. mutans* (11 mm) *C. albicans* (16 mm)	0.02 mL	[78]
*Ophiorrhiza trichocarpon* BI., *Ophiorrhza rugosa* and *Ophiorrhiza aff*. Nutans Cl. Ex Hk. f.	Alkaloids, coumarins, anthraquinone glycosides, scopoletin, saponins	Methanolic extract of the whole plant	Antibacterial activities of: *Ophiorrhiza trichocarpon BI.* *E.coli* (10 mm) *S.aureus* (10.5 mm) *Pseudomonas aeruginosa* (10 mm) *Ophiorrhza rugosa* *E.coli* (9.5 mm) *S.aureus* (10 mm) *Pseudomonas aeruginosa* (10 mm) *Ophiorrhiza aff*. Nutans Cl. Ex Hk. f. *E.coli* (10 mm) *S.aureus* (10.5 mm) *Pseudomonas aeruginosa* (10 mm)	75 mg/mL 10 mg/mL 10 mg/mL 100 mg/mL 10 mg/mL 50 mg/mL 100 mg/mL 10 mg/mL 10 mg/ml	[3]
*Ophiorrhiza shendurunii*	Lupan-20-ol-3(β)-yl hexadecanoate (51), lupan-20-ol-3(β)-yl acetate (52), olean-18-en-3(β)-yl hexadecanoate (53)	Hexane and chloroform extracts of the whole plant	Antibacterial activities of the chloroform extract using agar well diffusion assay are significant against: *B. subtilis* (22/18 mm) *E. coli* (19/23 mm). The hexane extract showed no antibacterial activity against *Escherichia coli* and *Bacillus subtilis.*	100 µL	[35]

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
