# Peer review of "Genus Ophiorrhiza: A Review of Its Distribution, Traditional Uses, Phytochemistry, Biological Activities and Propagation"

_molecules, 2020, doi:10.3390/molecules25112611_

Round 1
Reviewer 1 Report
The manuscript entitled ‘Genus Ophiorrhiza: A Review of Its Distribution, Traditional Uses, Phytochemistry, Biological Activities, and Propagation’ is a very interesting review about the plants of this genus. It is exhaustive, well written, and, in my opinion, fits with the journal scope. However, some limitations occur:
-INTRODUCTION should be more general, including phytochemistry and drug discovery process from natural substances. As it is, it could be another paragraph, entitled ‘Botany'.
-METHODOLOGY should be organized in a more systematic way. Some attempts have been done, indicating the sources and the inclusion criteria for the plants discussed in the paper. It should be interesting to indicate other inclusion and exclusion criteria and organize them in a PRISMA flow diagram.
-PHYTOCHEMISTRY. Adding some Structure-Activity Relationships to justify the biological activity showed by different phytochemicals could enrich the quality of the manuscript.
-Table 5. Please specify if doses refer to kg body weight or food. For the sake of completeness, the way and frequency of administration could be added.
-CONCLUSIONS seem to merely summarize the content of the work. It should be more critic, highlighting critical issues and strengths of new biotechnologies, as well as the sustainability of these plants as source of anticancer drugs from an environmental and economic point of view.
-in vivo and in vitro should be written in italic. Please check lines 62 (the) and 102 (juices).
Author Response
RESPOND TO REVIEWER 1 COMMENTS
The manuscript entitled ‘Genus Ophiorrhiza: A Review of Its Distribution, Traditional Uses, Phytochemistry, Biological Activities, and Propagation’ is a very interesting review about the plants of this genus. It is exhaustive, well written, and, in my opinion, fits with the journal scope. However, some limitations occur:
-INTRODUCTION should be more general, including phytochemistry and drug discovery process from natural substances. As it is, it could be another paragraph, entitled ‘Botany'.
>>>ANSWER: The introduction has been revised accordingly
-METHODOLOGY should be organized in a more systematic way. Some attempts have been done, indicating the sources and the inclusion criteria for the plants discussed in the paper. It should be interesting to indicate other inclusion and exclusion criteria and organize them in a PRISMA flow diagram.
>>>ANSWER: The methodology has been organised.
-PHYTOCHEMISTRY. Adding some Structure-Activity Relationships to justify the biological activity showed by different phytochemicals could enrich the quality of the manuscript.
>>>ANSWER: Thank you for the comments. The suggestion is very interesting, but , apologise that it is beyond our scope of review.
-Table 5. Please specify if doses refer to kg body weight or food. For the sake of completeness, the way and frequency of administration could be added.
>>>ANSWER: The Table has been modified
-CONCLUSIONS seem to merely summarize the content of the work. It should be more critic, highlighting critical issues and strengths of new biotechnologies, as well as the sustainability of these plants as source of anticancer drugs from an environmental and economic point of view.
>>>ANSWER: The conclusion has been revised.
-in vivo and in vitro should be written in italic. Please check lines 62 (the) and 102 (juices).
>>>ANSWER: They have been corrected thoroughly
Reviewer 2 Report
The contribution of Taher et al. entitled ‘Genus Ophiorrhiza: A Review of Its Distribution, 3 Traditional Uses, Phytochemistry, Biological Activities, and Propagation' attempts to cover almost 50 species of Ophiorrhiza plants, their use in traditional medicine, contain of phytochemicals that were identified in them, and to focus on few of them in context of biological activity. Such colossal task is rather well made however there are few things that in my opinion will bother potential reader and are thus diminishing tremendously value of this review. Below you can find (non-comprehensive) list of the things that must be changed before the acceptance of this manuscript. As I mentioned above, overall, this review is fine, comprehensive and well written, clearly defined scope that seems to be mostly fulfilled. However small but recurrent mistakes are largely diminishing its value. Therefore, I do propose the acceptance only after the major revisions.
List of things to change:
- Graphics
- inconsistent drawings with different size of letters (like oxygen, e.g. compound 53 59), bond lengths (53 vs 59 vs 60), overlapping atoms and lines (e.g. 28 and 34), stereochemistry not or not well defined (e.g. 35 and 36)
- carelessly drawn structures that are not clearly defined (39 – vinylic hydrogen atom; 46 – stereochemistry, carboxylic acid?...)
- not defined what Glu means
- charges – very often only cations or + sign in the middle of structure close to the hydrogen atom without any meaning; in some cases only cations or only anions are present and no counter cation/anion – have doubt that compounds were isolated as salts.
- Compound 21 use for bond “line” that is used to draw transition stateof the reaction
- In addition to graphic – the names of the compounds should be added under the corrected structures along with the number – it will simplify the reading. The same, when used name of the compound in the text, number should follow (completely missing since Biological activity chapter). Then it becomes very difficult to follow the text that speaks about e.g. camptothecin (page 16, line 277). How the structure looks like? Reader must go to Table 3 (alkaloids – where they come from) and there you find the name and the number that was attributed to it (11). Next go to Fig 2 to see structure that is unfortunately bizarrely drawn (why there are two Hydrogens on aromatic ring and others are not drawn?
- Thus
- Add names to Fig along with numbers
- All numbers should be in bold to be easily identifiable
- Add numbers to tables (Table 5 – no compound numbers )
- Add numbers of compounds along with their names into the text starting with chapter Biological activity and further
- Please re-check the trivial names of compounds – Ophiorrhiside A should be written instead of Ophiorrhisides A (in plural – see Table 3, last line)
- Page 7, line 146 – should be R1 and R2 (upper index) not R1 and R2
- All literature references are written in wrong format and should be rewritten in correct Molecules format.
Author Response
RESPOND TO REVIEWER 2 COMMENTS
The contribution of Taher et al. entitled ‘Genus Ophiorrhiza: A Review of Its Distribution, 3 Traditional Uses, Phytochemistry, Biological Activities, and Propagation' attempts to cover almost 50 species of Ophiorrhiza plants, their use in traditional medicine, contain of phytochemicals that were identified in them, and to focus on few of them in context of biological activity. Such colossal task is rather well made however there are few things that in my opinion will bother potential reader and are thus diminishing tremendously value of this review. Below you can find (non-comprehensive) list of the things that must be changed before the acceptance of this manuscript. As I mentioned above, overall, this review is fine, comprehensive and well written, clearly defined scope that seems to be mostly fulfilled. However small but recurrent mistakes are largely diminishing its value. Therefore, I do propose the acceptance only after the major revisions.
List of things to change:
- Graphics
- inconsistent drawings with different size of letters (like oxygen, e.g. compound 5359), bond lengths (53 vs 59 vs 60), overlapping atoms and lines (e.g. 28 and 34), stereochemistry not or not well defined (e.g. 35 and 36)
>>>ANSWER: The structures have been revised accordingly
- carelessly drawn structures that are not clearly defined (39 – vinylic hydrogen atom; 46 – stereochemistry, carboxylic acid?...)
>>>ANSWER: The structures have been revised accordingly
- not defined what Glumeans
- >>>ANSWER: The structures have been revised accordingly (Glu(s) have been changed with Glc
- charges– very often only cations or + sign in the middle of structure close to the hydrogen atom without any meaning; in some cases only cations or only anions are present and no counter cation/anion – have doubt that compounds were isolated as salts.
- >>>ANSWER: They were drawn according to original sources.
- Compound 21 use for bond “line” that is used to draw transition stateof the reaction
- >>>ANSWER: The structure has been revised
- In addition to graphic – the names of the compounds should be added under the corrected structures along with the number – it will simplify the reading. The same, when used name of the compound in the text, number should follow (completely missing since Biological activity chapter). Then it becomes very difficult to follow the text that speaks about e.g. camptothecin (page 16, line 277). How the structure looks like? Reader must go to Table 3 (alkaloids – where they come from) and there you find the name and the number that was attributed to it (11). Next go to Fig 2 to see structure that is unfortunately bizarrely drawn (why there are two Hydrogens on aromatic ring and others are not drawn?
- Thus
- Add names to Fig along with numbers
>>>ANSWER: The compunds have been named and numbered
- All numbers should be in bold to be easily identifiable
>>>ANSWER: The numbers have been bolded
- Add numbers to tables (Table 5 – no compound numbers )
>>>ANSWER: The compound have been numbered
- Add numbers of compounds along with their names into the text starting with chapter Biological activity and further
>>>ANSWER: The compounds have been labeled with number and name
- Please re-check the trivial names of compounds – Ophiorrhiside A should be written instead of OphiorrhisidesA (in plural – see Table 3, last line)
>>>ANSWER: Corrected
- Page 7, line 146 – should be R1and R2 (upper index) not R1 and R2
>>>ANSWER: Revised
- All literature references are written in wrong format and should be rewritten in correct Molecules format.
>>>ANSWER: The references have been edited according to journal style.
Round 2
Reviewer 1 Report
none
Author Response
Thank you
Nothing to reply
Reviewer 2 Report
Overall I think that corrections increased the value of manuscript, however there are several issues that must be addressed before the acceptance of the review:
1) structure drawings:
stereochemistry of compounds 4, 5, 9, 10, 15, 16
drawing of CO2H (bonded via hydrogen atom?)
signes "+" nearby the H atoms in 18, 19
comp. 21, 22 - CH2OH group is equatorial or axial? in the sugar structure?
24,25 confusing overlap of atoms
32 - olefin is in E or Z configuration?
48, 49 / Glu in structure but Glc in explication....
2) but most importantly - as mentioned before some structures are presented as cations or even double cations. Answer that original litt is showing them as those is unfortunately not relevant (could not check - do not have access), because if it is the case then it is putting any result in that particular paper in doubt (especially in context of the biological activity of the given molecule) and therefore it should not be included in this review (misleading info). Please could you check up again and if unclear then see with authors. Another option would be exclusion of these compounds out of the review.
Author Response
Response to the reviewer comments
Comments and Suggestions for Authors
Dear reviewer,
Thank you for your very valuable comments, kindly refer to our respond:
Comment:
Overall I think that corrections increased the value of manuscript, however there are several issues that must be addressed before the acceptance of the review:
1) structure drawings:
stereochemistry of compounds 4, 5, 9, 10, 15, 16
drawing of CO2H (bonded via hydrogen atom?)
Ans: >>>The structures have been corrected. Thank you
signes "+" nearby the H atoms in 18, 19
Ans:>>>The structures have been corrected
comp. 21, 22 - CH2OH group is equatorial or axial? in the sugar structure?
Ans:>>>The structure has been confirmed as equatorial position
24,25 confusing overlap of atoms
Ans:>>>The structures have been improved.
32 - olefin is in E or Z configuration?
Ans:>>>The structure has been corrected as E configuration.
48, 49 / Glu in structure but Glc in explication....
Ans:>>>The structures have been revised
2) but most importantly - as mentioned before some structures are presented as cations or even double cations. Answer that original litt is showing them as those is unfortunately not relevant (could not check - do not have access), because if it is the case then it is putting any result in that particular paper in doubt (especially in context of the biological activity of the given molecule) and therefore it should not be included in this review (misleading info). Please could you check up again and if unclear then see with authors. Another option would be exclusion of these compounds out of the review.
>>>Thank you very much for the constructive comment. We realised that there is no meaning of the cations nearby the Hydrogen atom. We mistakenly drew the structure with the cation symbol, which were used as a proposed intermediate compound in hydrolytic degradation of beta carboline alkaloid. However the structure have been revised.
Best regards,
Dr. Muhammad Taher